# Do Deep Regional Trade Agreements Improve Residents’ Health? A Cross-Country Study

**DOI:** 10.3390/ijerph192114409

**Published:** 2022-11-03

**Authors:** Zhizhong Liu, Qianying Chen, Guangyue Liu, Xu Han

**Affiliations:** 1School of Finance and Trade, Liaoning University, Shenyang 110036, China; 2School of Marxism, Liaoning University, Shenyang 110036, China

**Keywords:** regional trade agreements (RTAs), depth, residents’ health, employment effects, environmental effects

## Abstract

The development trend of deepening regional trade agreements (RTAs) is becoming more prominent, traditional RTAs based on border terms continue to shift to deep RTAs based on the high level of border terms and a series of post-border terms, but the relationship between deep RTAs and residents’ health has not drawn much attention. Based on Gallup World Poll data from 2009 to 2017 covering 786,040 respondents in 143 countries, this study empirically examined the impact of deep RTAs on the health of residents as well as its influence mechanisms by using the combination of fixed effects and stepwise regression. The results show that deep RTAs have a significantly positive impact on residents’ health, which means that an increase in the depth of RTAs can improve residents’ health. However, the impact of deep RTAs on residents’ health is heterogeneous, caused by the different terms of RTAs, the different income levels of different countries, and the different types of residents. Meanwhile, deep RTAs mainly improve the health of residents through employment effects and environmental effects. This study highlights the importance of deep RTAs for improving the health of residents and provides new ideas for governments to assist in the formulation of policies that can effectively improve their residents’ health.

## 1. Introduction

In recent years, great changes have taken place in regional trade agreements (RTAs). On one hand, the number of RTAs has escalated. According to the WTO Regional Trade Agreements Database, the number of RTAs worldwide has increased from 50 in 1990 to 355 in 2021, and the coverage of RTAs is also expanding. On the other hand, the content of RTAs is being enriched. Compared with traditional RTAs, the new generation of agreements has a higher level of liberalization with regard to border measures, such as tariff and non-tariff barriers, and pays more attention to policy areas behind borders, such as labor market supervision and environmental protection. RTAs continue to extend from a single product, department, and field to a broader range of trade, economic, and social areas. As an essential carrier to promoting regional economic development, deep RTAs have had an important impact on the economic and social development of countries, especially on the health of the residents within those countries. RTAs can not only improve the output of export enterprises and increase the employment willingness of workers, so as to improve residents’ health through employment effects, but they also can promote clean production and the import of hygienic products in enterprises, so as to improve residents’ health through environmental effects. In addition, as an essential human capital, health is crucial to a country’s economic growth and social development [1], and it has always been the primary social issue that governments continue to focus on. Therefore, an in-depth analysis of the impact of deep RTAs on residents’ health is not only conducive to promoting the deep RTAs of countries, but is also beneficial for strengthening healthy human capital, so as to better promote economic and social development.

At present, the trade effects of deep RTAs, the factors affecting residents’ health, and the impact of trade liberalization on residents’ health are the three main aspects that have been researched by scholars. First, regarding the trade effects of deep RTAs, scholars have mainly focused on the overall depth and the depth of the terms. Some scholars have studied the trade effects of the overall depth of RTAs and demonstrated that an increase in the depth of RTAs has a positive impact on traditional trade [2,3], value chain trade [4,5,6,7], and the value chain division of labor [8,9]. Meanwhile, some scholars have researched the trade effects of the depth of the terms in RTAs and examined the heterogeneous impacts of the depth of environmental terms [10,11], the depth of digital trade facilitation terms [12], the depth of intellectual property terms [13], and the depth of technical trade measures terms [14] on a country’s foreign trade. Second, regarding the factors affecting residents’ health, scholars have mainly analyzed the internal and external factors that affect residents’ health. Some scholars have proposed that environmental pollution [15,16], urbanization [17], the internet [18], road construction [19], and foreign direct investment [20,21,22] all have an important impact on residents’ health. Other scholars have argued that internal factors such as employment [23], income [24], education level [25], and the medical and health services enjoyed by residents [26] are all important for residents’ health, and significantly promote improvements in residents’ health. Third, regarding the impact of trade liberalization on residents’ health, scholars hold two different views. Some scholars have argued that trade liberalization could effectively improve residents’ health through the channels of knowledge spillover [27], economic growth [28], public health [29], and income [30]. While others have argued that trade liberalization has worsened residents’ health by extending workers’ working hours [31] or increasing environmental pollution [32]. Furthermore, few scholars have analyzed the impact of traditional RTAs on residents’ health. Some scholars have analyzed the adverse impacts of traditional RTAs on residents’ health from a qualitative perspective [33,34]. Others have taken the RTAs signed by the European Union and the United States as samples and demonstrated the positive impacts of traditional RTAs on residents’ health by using the dummy variables method [35,36].

From the above analysis, it can be found that there have been abundant studies on the trade effects of deep RTAs, the factors affecting residents’ health, and the impact of trade liberalization on residents’ health. Meanwhile, some scholars have used qualitative analysis methods or dummy variable methods to analyze the impact of RTAs on residents’ health. For example, Stiglitz [34] qualitatively analyzed the negative effects of traditional RTAs on the health of residents in developing countries; Venkatamaran & Stevens [35] analyzed the positive effects of traditional FTAs on health outcomes using the dummy variable method based on the sample of countries that had signed FTAs with the United States or the European Union, etc. However, there are no studies examining the impact of deep RTAs signed around the world on residents’ health from the perspective of terms depth, and there are also no studies deeply analyzing the mechanisms by which deep free trade agreements affect residents’ health. Compared with existing studies, the contributions of this study are as follows. First, this study focuses on deep RTAs, covering 18 terms and 1028 sub-terms, and quantifies them through the “terms counting index” method, which avoids the deficiencies of the dummy variable method and reflects the differences among RTAs. Second, this study uses Gallup World Poll data from 2009 to 2017 covering 786,040 respondents in 143 countries around the world to empirically examine the impact of deep RTAs on residents’ health, and the heterogeneous impacts caused by the different terms of RTAs, the different income levels of different countries, and the different types of residents. Third, this study analyzes and examines the mechanisms of deep trade agreements and their effect on residents’ health; that is, deep RTAs mainly improve residents’ health through employment effects and environmental effects.

## 2. Theoretical Analysis and Hypothesis

Compared with traditional RTAs, deep RTAs cover a more comprehensive range of policy areas, including border measures, such as tariffs and non-tariff barriers, as well as environmental protection, labor market supervision, competition policies, services, etc. Mattoo et al. [37] found that deep RTAs not only cover all the provisions within the scope of the current World Trade Organization (WTO) regulations, but also include some important provisions beyond the scope of the WTO regulations. Therefore, deep RTAs can effectively increase domestic employment and reduce domestic environmental pollution, thereby improving residents’ health through employment effects and environmental effects. Based on this, the first hypothesis is proposed:

**Hypothesis** **1.***Deep regional trade agreements can promote an improvement in residents’ health*.

First, deep RTAs can affect residents’ health through employment effects. On one hand, border measures such as tariffs and non-tariff barriers that are included in deep RTAs can effectively reduce the trade costs that enterprises need to pay for export, including fixed trade costs such as market entry and information exchange, etc., as well as variable trade costs such as product transportation and trial communication, etc., and reduce trade diversion while promoting trade creation [3]. This not only enhances the growth ability of enterprises through learning effects and knowledge spillover effects, and promotes the output expansion of enterprises, but can also encourage more domestic enterprises to enter the export market, thereby providing more jobs in the country. Meanwhile, the behind-border measures, such as labor market supervision, included in deep RTAs can maximize the protection of workers’ rights and interests, and effectively improve workers’ willingness to be employed. Therefore, deep RTAs can increase domestic employment and promote an increase in the domestic labor force participation rate [38,39,40]. On the other hand, outstanding employment performances can improve residents’ health by affecting their income, health care security, and mental stress. From the perspective of income, outstanding employment performances can increase residents’ income and allow them to have more money for dietary adjustment, health care expenditure and specialized health investments such as: travel, vacations, fitness, exercise, etc. [24,41]. From the perspective of health care, outstanding employment performances can provide residents with more publicly funded health inspections, reducing the pressure on residents’ health care expenditure. From the perspective of mental stress, outstanding employment performances can relieve the anxiety of residents caused by unemployment and the psychological pressure caused by economic burden, thereby improving the life satisfaction of residents and improve the subjective health level of residents’ self-assessment [42,43]. Based on the above analysis, the second hypothesis is proposed:

**Hypothesis** **2.***Deep regional trade agreements improve residents’ health through employment effects*.

Second, deep RTAs can affect residents’ health through environmental effects. On one hand, deep RTAs contain a complete and systematic environmental protection framework, which covers 8 secondary terms and 55 tertiary terms related to environmental protection, regulating the environmental protection behavior of each member state from various aspects, such as environmental goals, balance between environmental and trade goals, enforcement mechanisms, external assistance, general environmental protection areas, participation in promoting environmental objectives, etc. Therefore, deep RTAs can form a kind of environmental regulation for domestic enterprises in each country. The environmental regulation will force enterprises to shift their production processes to cleaner ones and force them to abandon the production of polluting products [44]. This greatly promotes cleaner production of enterprises and the import of cleaner intermediate products, which ultimately reduces pollutants that are harmful to residents’ health, such as PM2.5, and improves domestic environmental quality. On the other hand, environmental pollution is an important factor affecting residents’ health. The adverse impact of environmental pollution on residents’ health is second only to residents’ age [45,46]. The aggravation caused by environmental pollution not only directly accelerates the decline in individual residents’ health, but also indirectly affects their health output through other factors, such as personal exercise and food intake. Meanwhile, environmental pollution will increase the health expenditure of residents [47], which will indirectly reduce the income of residents and increase the stress of their lives, and eventually leads to the decline of residents’ health. Therefore, an improvement in environmental quality can effectively slow down the decline in residents’ health, directly caused by environmental pollution, avoid health output decline, indirectly caused by environmental pollution, and reduce the pressure on residents’ health expenses, creating a healthy living environment for residents to improve their health. Based on the above analysis, the third hypothesis is proposed:

**Hypothesis** **3.***Deep regional trade agreements improve residents’ health through environmental effects*.

## 3. Materials and Methods

### 3.1. Variables

#### 3.1.1. Dependent Variable

The dependent variable is residents’ health, which is measured by the personal health index in the Gallup World Poll Database, and ranges from 0 to 100. Meanwhile, the residents’ health value in other countries, except the Arab countries, is measured at an interval of 20, and a larger value indicates a higher level of residents’ health. To be specific, the personal health index is a subjective measure, which represent the perceptions of one’s own health, both physical and mental and incidence of pain, sadness, and worry. Additionally, the personal health index is measured based on the Gallup questionnaire, which includes a standard set of questions related to personal health. For example, have you tried to get any information about medicine, disease, or health in the past 30 days? Would you like to know more about medicine, disease, or health? How much do you trust medical and health advice from medical workers, such as doctors and nurses, in this country? etc., and the results of Gallup questionnaire can represent more than 98% of the world’s adult population.

#### 3.1.2. Independent Variable

The key independent variable is the depth of RTAs, and the specific calculation method is as follows: First, this study mainly considers 18 terms, including 1028 sub-terms, when measuring the depth of RTAs [37]. The 18 terms are anti-dumping, countervailing duties, export taxes, competition policies, investment, intellectual property policies, services, trade facilitation, state subsidies, labor market, migration, movement of capitals, public procurement, rules of origin, environment, sanitary and phytosanitary measures (SPS), state trading enterprise (STE), and technical barriers to trade (TBT). Second, this study uses the “terms counting index” method [48] to score the 18 terms. If a regional trade agreement contains a sub-term among the 18 terms, the corresponding term variable is assigned a value of “1”; otherwise, it is “0”. This study then uses the vertical aggregation method to aggregate the scores of 1028 sub-terms in each regional trade agreement to obtain the depth of each agreement. Finally, this study aggregates and normalizes the depth of all RTAs signed by countries, and then derives the variable of the depth of RTAs. A larger depth of RTAs value indicates a deeper regional trade agreement. The scores of various terms of RTAs are shown in Table 1.

#### 3.1.3. Control Variables

The control variables include individual-level control variables and country-level control variables. The control variables at the individual level specifically include: residents’ income, measured by a total annual household net income logarithm and calculated with the current US dollar as the base period; and residents’ education level. If the resident has completed basic education, a value of “1” will be assigned. If the resident has completed vocational education, a value of “2” will be assigned. If the resident has completed higher education, a value of “3” will be assigned. Residents’ age is measured by the actual age of the resident. Regarding residents’ gender, if the resident is male, a value of “1” will be assigned; if the resident is female, a value of “0” will be assigned. For the marital status of residents, if the resident is married, which includes married and domestic partners, a value of “1” will be assigned; if the resident is unmarried, which includes single, divorced, and widowed individuals, a value of “0” will be assigned. With regard to residents’ residential area, if the resident lives in a city or suburb, a value of “1” will be assigned; if the resident lives in a rural area or small town, a value of “0” will be assigned. The control variables at the national level include: GDP per capita, calculated with the constant 2015 US dollar as the base period; health care expenditure per capita, measured by purchasing power parity and calculated with the current US dollar as the base period; and foreign trade dependence, measured by the proportion of import and export trade in the gross national product. In addition, this study includes two mediating variables, employment and environmental pollution. Employment is represented by the labor force participation rate, and environmental pollution is represented by the average annual PM2.5 exposure.

### 3.2. Data Sources and Statistical Characteristics

The data used in this study were mainly derived from the Gallup World Poll Database, the World Bank Trade Agreement Content Database (https://datacatalog.worldbank.org/search/dataset/0039575 (accessed on 5 July 2022)), and the World Bank Development Indicators Database (https://databank.worldbank.org/source/world-development-indicators (accessed on 16 July 2022)). The sample period ranges from 2009 to 2017, and the sample covers 786,040 residents in 143 countries, mainly including some important developed countries, such as the United States, the United Kingdom, Germany, France, Japan, and some important developing countries, such as China, Malaysia, Brazil, India, and Russian Federation. To be specific, the residents’ health data were sourced from Gallup World Poll Database, which can be obtained by submitting a request to the Gallup; the depth of RTAs data were sourced from the World Bank Trade Agreement Content Database; individual-level control variables data, such as residents’ income, education level, age, gender, marital status, and residential area, were all sourced from the Gallup World Poll Database; national-level control variables data, such as GDP per capita, health care expenditure per capita, and foreign trade dependence, were all sourced from the World Bank Development Indicators Database; and the annual average exposure to PM2.5 data and the labor force participation rate data were sourced from the World Bank Development Indicators Database. The information about the symbols, definitions, and data sources of the independent, dependent, and control variables are all presented in Table 2.

The descriptive statistics of the variables are presented in Table 3. Among them, the minimum residents’ health index value is 0, the maximum residents’ health index value is 100, and the average residents’ health index value is 70.2256. The minimum depth of RTAs value is −1.5423, the maximum depth of RTAs value is 1.6276, and the average depth of RTAs value is −0.5487, indicating that the depth of RTAs signed by countries is acceptable, but there is still room for improvement. The above conclusions are only preliminary judgments and have not been obtained through rigorous hypothesis testing. However, these results reveal some generalized characteristics of residents’ health and depth of RTAs.

### 3.3. Model Specification

#### 3.3.1. Benchmark Regression Model

Following the studies of Venkatamaran & Stevens [35] and Barlow et al. [36], and considering the research dimension of this paper at the individual-country level, the following econometric model is constructed to investigate the impact of deep RTAs on residents’ health: (1)$${\mathrm{PHI}}_{\mathrm{ijt}}{=\beta }_{0}+\beta_{1}{\mathrm{Depth}}_{\mathrm{jt}}+{\beta X+\lambda }_{j}+\lambda_{t}+\lambda_{j}\times {t+\mu }_{\mathrm{ijt}}$$
where i represents the individual; j is the country; t denotes the year; ${\;\mathrm{PHI}}_{\mathrm{ijt}}$ stands for the health level of resident i of country j in year t; ${\;\mathrm{Depth}}_{\mathrm{jt}}$ represents the depth of RTAs of country j in year t, which is the key independent variable of this study; and X is the set of control variables, including individual-level control variables and country-level control variables. The control variables at the individual level specifically include: residents’ income (${\mathrm{Lnincome}}_{\mathrm{it}}$), residents’ education level (${\mathrm{Education}}_{\mathrm{it}}$), residents’ age (${\mathrm{Age}}_{\mathrm{it}}$
), residents’ gender (${\mathrm{Male}}_{\mathrm{it}}$), residents’ marital status (${\mathrm{Marriage}}_{\mathrm{it}}$), and residents’ residential area (${\mathrm{Urban}}_{\mathrm{it}}$). The control variables at the national level include: GDP per capita (${\mathrm{LnGDPpc}}_{\mathrm{jt}}$), health care expenditure per capita (${\mathrm{Lnhepc}}_{\mathrm{jt}}$), and foreign trade dependence (${\mathrm{FTD}}_{\mathrm{jt}}$). $\lambda_{j}$ is the country fixed effect; $\lambda_{t}$
is the year fixed effect; $\lambda_{j}\times t$ is the country linear time trend, which is used to reduce the bias caused by omitted variables at the country macro level; and $\mu_{\mathrm{ijt}}$ is the random error term.

#### 3.3.2. Mechanism Analysis Models

To further test the channel role of employment effects and environmental effects in the process of deep RTAs affecting residents’ health, this study constructed the following models: (2)$${\mathrm{Laborpar}}_{\mathrm{jt}}=b_{0}+b_{1}{\mathrm{Depth}}_{\mathrm{jt}}+{\mathrm{bX}+\lambda }_{j}+\lambda_{t}+\lambda_{j}\times {t+\varepsilon }_{\mathrm{jt}}$$
(3)$${\mathrm{Pollution}}_{\mathrm{jt}}=\alpha_{0}+\alpha_{1}{\mathrm{Depth}}_{\mathrm{jt}}+{\alpha X+\lambda }_{j}+\lambda_{t}+\lambda_{j}\times {t+\tau }_{\mathrm{jt}}$$
(4)$${\;\mathrm{PHI}}_{\mathrm{ijt}}=d_{0}+d_{1}{\mathrm{Depth}}_{\mathrm{jt}}+d_{2}{\mathrm{Laborpar}}_{\mathrm{jt}}+d_{3}{\mathrm{Pollution}}_{\mathrm{jt}}+{\mathrm{dX}+\lambda }_{j}+\lambda_{t}+\lambda_{j}\times {t+\chi }_{\mathrm{ijt}}$$
where Model (2) examines the impact of deep RTAs on the labor force participation rate; Model (3) examines the impact of deep RTAs on the average annual exposure of PM2.5; Model (4) examines the impact of three variables on residents’ health, and the three variables are the depth of RTAs, the labor force participation rate and the average annual exposure of PM2.5; ${\mathrm{Laborpar}}_{\mathrm{jt}}$ represents the labor force participation rate; ${\mathrm{Pollution}}_{\mathrm{jt}}$ represents the average annual exposure of PM2.5; $\varepsilon_{\mathrm{jt}}$, $\tau_{\mathrm{jt}}$, $\chi_{\mathrm{ijt}}$ are random error terms; and the meanings of other variables are the same as those of Model (1).

## 4. Results and Discussions

### 4.1. Benchmark Empirical

Table 4 shows the benchmark empirical results of the impact of deep RTAs on residents’ health. Column (1) only considers the variable of the depth of RTAs. Columns (2) and (3) gradually include the individual-level control and country-level control variables. It can be seen from Table 4 that the estimated coefficients of the depth of RTAs are all significantly positive at the 1% level, and the coefficient of the depth of RTAs in the Column (3) is 1.987, which means that a one-point increase in the depth of RTAs will improve residents’ health by 1.987 points. Meanwhile, with the addition of the relevant control variables, the sign and significance of the estimated coefficients of the depth of RTAs have not changed, indicating that an increase in the depth of RTAs signed by a country can promote improvements in the health of domestic residents, which is because deep RTAs can not only increase the labor force participation rate within countries, but can also reduce domestic air pollution, thus improving residents’ health through employment effects and environmental effects. The above conclusions not only verify Hypothesis 1, but also provide new ideas for countries to improve residents’ health under the development trend of deepening RTAs. 

In terms of control variables at the individual level, the estimated coefficient of residents’ income is significantly positive, indicating that an increase in residents’ income can improve residents’ health; the estimated coefficient of residents’ education level is significantly positive, indicating that an improvement in residents’ education level can improve residents’ health; the estimated coefficient of residents’ age is significantly negative, indicating that the level of residents’ health declines with an increase in residents’ age; the estimated coefficient of residents’ gender is significantly positive, indicating that male residents have better health than female residents; the estimated coefficient of residents’ marital status is significantly positive, indicating that married residents have better health than unmarried residents, which is mainly because the families of married residents are more resistant to risks; and the estimated coefficient of residents’ residential area is significantly negative, indicating that, affected by factors such as living costs and work pressure, the health level of urban and suburban residents is significantly lower than that of rural and small-town residents.

In terms of control variables at the country level, the estimated coefficient of GDP per capita is positive but not significant, which is consistent with the findings of Novignon [49], and is mainly due to the impact of government spending efficiency; the estimated coefficient of health care expenditure per capita is significantly positive, indicating that an increase in expenditure on health care in a country can improve residents’ health; and the estimated coefficient of foreign trade dependence is significantly positive, indicating that a higher level of foreign trade dependence can promote an improvement in residents’ health. This is mainly because countries with a high dependence on foreign trade can improve their domestic medical status by introducing advanced medical equipment and medical technology.

### 4.2. Robustness Test

To ensure the robustness of the benchmark results, this study conducted a series of robustness tests by replacing the key independent variable, performing pseudo-panel regression, using the ordered logit method to regress, and expanding the sample interval.

#### 4.2.1. Replacing the Depth of RTAs

As the number of RTAs signed by each country varies, this study uses the arithmetic average method to recalculate the depth of RTAs as follows: divide the sum of the depth of RTAs signed by countries in that year by the number of RTAs signed by countries in that year, then normalize it, thereby obtaining the average depth of RTAs. Column (1) of Table 5 shows the regression results of replacing the depth of RTAs, and the results show that the estimated coefficient of Averagedepth is significantly positive, indicating that the conclusions of this study can be considered robust.

#### 4.2.2. Pseudo-Panel Regression

As the respondents of the Gallup Poll are not the same every year, this study adds up the individual characteristic variables and constructs pseudo-panel data, including country characteristic variables and individual characteristic variables. Among them: residents’ health index, age, and income are all measured by their mean; residents’ gender is measured by the proportion of male respondents in the country; residents’ education level is measured by the proportion of the country’s respondents with higher education qualifications; residents’ residential area is measured by the proportion of urban or suburban population in the country’s respondents; and residents’ marital status is measured by the proportion of the country’s respondents who are married. This study then adopts the least squares dummy variable method to obtain an estimate. Column (2) of Table 5 shows the regression results of performing pseudo-panel regression, and the results show that the depth of RTAs significantly promotes an improvement in residents’ health, indicating that the conclusions of this study are robust.

#### 4.2.3. Regression Using the Ordered Logit Method

As the PHI index, which measures residents’ health, is an ordinal variable, this study uses the ordered logit method to re-regress Model (1). Column (3) of Table 5 shows the regression results of using the ordered logit method to regress, and the results show that the estimated coefficient of the depth of RTAs is still significantly positive, indicating that the conclusions of this study are valid after changing the method of regression.

#### 4.2.4. Expand the Sample Interval

As the age range of the sample residents in this study is 25–64 years old, in order to further investigate the impact of the depth of RTAs on the health of residents who are 18–24 years old, the original sample was included in the sample of residents aged 18–24 years. Column (4) of Table 5 shows the regression results of expanding the sample interval, and the results show that the depth of RTAs has a significantly positive impact on residents’ health, indicating that the conclusions of this study are robust.

### 4.3. Endogeneity Test

This study addresses the endogenous estimation bias caused by omitted variables by controlling the country fixed effect, the year fixed effect, and the country linear time trend; however, there may be a two-way causal relationship between deep RTAs and residents’ health. On the one hand, deep RTAs can promote the improvement of residents’ health through employment effects and environmental effects; On the other hand, the improvement of residents’ health will restrict signing of RTAs and encourage countries to sign deep RTAs that include more behind-border terms, such as labor market and environment terms. Therefore, this study uses the sum of the depth of RTAs signed by each country’s trade agreement partners (${\mathrm{Depth}}^{\mathrm{IV}}$) as an instrumental variable to re-regress. The main reasons are as follows: First, there is a natural relationship between the sum of the depth of RTAs signed by each country’s trade agreement partners and the depth of each country’s RTAs, which satisfies the correlation assumption of the instrumental variable. Second, the sum of the depth of RTAs signed by each country’s trade agreement partners will not have a direct impact on the residents’ health of each country, which satisfies the exogenous assumption of the instrumental variable.

Table 6 shows the regression results using the instrumental variable method. The results show that the instrumental variable has passed the Kleibergen–Paap rk LM test and the Kleibergen–Paap rk Wald F test, indicating that the instrumental variable selected in this study is reasonable and effective. Meanwhile, compared with the benchmark empirical results in Table 4 the sign and significance of the estimated coefficients of ${\mathrm{Depth}}^{\mathrm{IV}}$ have not changed, and with the addition of individual-level and country-level control variables, the estimated coefficients of
${\mathrm{Depth}}^{\mathrm{IV}}$ are all significant at the 1% level, indicating that an increase in the depth of RTAs signed by a country can improve residents’ health. The result also strongly verifies that, after addressing the problem of two-way causality, the conclusions of this study are still valid.

### 4.4. Mechanism Analysis

Table 7 shows the mechanism test results. It can be seen from columns (2) and (3) of Table 7 that the impact of the depth of RTAs on the labor force participation rate is significantly positive, and the impact of the depth of RTAs on the average annual exposure to PM2.5 is significantly negative. Meanwhile, it can be seen from column (4) of Table 7 that the estimated coefficient of labor force participation rate is significantly positive, and the estimated coefficient of average annual exposure to PM2.5 is significantly negative, indicating that an increase in the labor force participation rate and a reduction in the average annual exposure to PM2.5 can both improve residents’ health. Therefore, the mediating effect in this study is established; that is, deep RTAs mainly improve residents’ health through employment and environmental effects, which verifies Hypothesis 2 and Hypothesis 3.

### 4.5. Heterogeneity Analysis

As the impact of deep RTAs on residents’ health may be heterogeneous due to the different terms of RTAs, the different income levels of different countries, and the different types of residents, this study conducted heterogeneity analyses.

#### 4.5.1. Based on Different Terms of RTAs

Different depths of terms in RTAs may have heterogeneous impacts on residents’ health. Based on this, this study divided the depth of RTAs into the depth of “WTO+” terms (LnWTO+) and the depth of “WTO-X” terms (LnWTO+) to estimate separately. Among them, “WTO+” terms refer to terms within the scope of the current WTO regulations, including 10 terms and 505 sub-terms. The 10 terms are: anti-dumping, countervailing duties, export taxes, state subsidies, public procurement, rules of origin, trade facilitation, SPS, TBT, and STE. The “WTO-X” terms refer to terms outside the scope of the WTO regulations, including 8 terms and 523 sub-terms. The 8 terms are competition policy, investment, intellectual property policy, services, labor market, migration, environment, and movement of capital. Columns (1) and (2) of Table 8 show the results of the heterogeneity regression of differentiating the terms of RTAs, and the results show that the depth of the “WTO+” terms and the depth of the “WTO-X” terms significantly improved residents’ health, but the depth of the “WTO+” terms have a more significant impact. This is mainly because the RTAs signed by each country include a number of “WTO+” terms and compared with the “WTO-X” terms, the coverage rate of “WTO+” terms in each regional trade agreement are higher. Therefore, the depth of the “WTO+” terms can better promote an improvement in residents’ health.

#### 4.5.2. Based on Different Income Levels of Different Countries

Deep RTAs may have heterogeneous impacts on the health of residents in countries with different income levels. Based on this, this study divided the sample countries into high-income countries and low-income or middle-income countries, according to the World Bank’s country income classification standard. Columns (3) and (4) of Table 8 display the results of the heterogeneity regression of differentiating the income levels of different countries, and the results show that the depth of RTAs significantly improved the health of residents in low-income and middle-income countries but had no significant impact on the health of residents in high-income countries. This is mainly because, compared with high-income countries, the domestic employment mechanism of low-income and middle-income countries still needs to be improved, and environmental protection terms are relatively lacking. Therefore, signing deep RTAs can not only effectively increase the domestic labor force participation rate, but can also reduce domestic air pollution, thus promoting an improvement in residents’ health through employment effects and environmental effects.

#### 4.5.3. Based on Different Types of Residents

The first heterogeneity analysis is based on the residents in different residential areas. The sample residents are divided into urban residents and rural residents according to their living areas. Among them, the urban residents include residents living in cities or suburbs, and the rural residents include residents living in rural areas or small towns. Columns (1) and (2) of Table 9 show the results of the heterogeneity regression of differentiating residents’ residential areas. The results show that the depth of RTAs can significantly improve the health of both urban and rural residents, but it has a more significant impact on the health of rural residents. This is mainly because, compared with urban areas, there are few jobs and opportunities in rural areas, and the employment status of rural residents is unsatisfactory. The signing of deep RTAs will promote the employment of rural residents, and then greatly improve the health of rural residents through the employment effect. In addition, although rural areas are less polluted than urban areas, the signing of deep RTAs will further improve the overall environmental level of rural areas, and then slightly improve the health status of rural residents through environmental effects. 

The second heterogeneity analysis is based on the residents with different education levels. The sample residents are divided into higher education residents, basic education residents, or vocational education residents, according to their level of education. Columns (3) and (4) of Table 9 show the results of the heterogeneity regression of differentiating residents’ education levels, and the results show that the impact of the depth of RTAs on the health level of residents with a basic education and a vocational education is significantly positive, and the impact on the health level of residents with a higher education is also positive, but not significant. This is mainly because, compared with higher education residents, the employment status of basic education residents and vocational education residents is more uncertain, and their living environments are mostly at the medium level or below; therefore, the employment effects and the environmental effects generated by deep RTAs can effectively improve their employment status and living environment, thereby improving their health.

## 5. Conclusions

Based on Gallup World Poll data from 2009 to 2017 covering 786,040 respondents in 143 countries, this study empirically examined the impact of deep regional trade agreements on residents’ health and its influence mechanisms. The results are as follows: First, deep RTAs have a significant positive impact on residents’ health, which means that an increase in the depth of RTAs can improve residents’ health. Second, the impact of deep RTAs on residents’ health is heterogeneous due to the different terms of RTAs, the different income levels of different countries, and the different types of residents. To be specific: the depth of the “WTO+” terms and the “WTO-X” terms significantly improve residents’ health, but the depth of the “WTO+” terms has a more significant impact; the depth of RTAs significantly improves the health of residents in low-income and middle-income countries, but has no significant impact on the health of residents in high-income countries; the depth of RTAs can significantly improve the health of both urban and rural residents, but it has a more significant impact on the health of rural residents; the impact of the depth of RTAs on the health level of residents with a basic education or a vocational education is significantly positive, and the impact on the health level of residents with a higher education is also positive, but not significant. Third, deep RTAs mainly improve residents’ health through employment effects and environmental effects. Deep RTAs can not only improve residents’ health by increasing domestic employment but can also improve residents’ health by reducing domestic environmental pollution. 

There are several policy implications for the findings of this study. First, all countries should be committed to increasing the depth of their RTAs, and continue to promote the deepening development of RTAs, so as to maximize the potential role of deep RTAs in improving residents’ health. On one hand, countries should continue to expand the coverage of RTAs, and actively conduct regional trade agreement negotiations with potential partners. Furthermore, on the basis of strengthening trade cooperation in traditional policy areas such as tariffs and non-tariff barriers, new areas should be further strengthened, such as the environment and labor. On the other hand, countries should upgrade the RTAs they have already signed. They should not only actively promote a high degree of liberalization in the border policy areas, but should also actively incorporate the behind-border terms, such as environmental protection and labor supervision, In addition, countries should grasp the concept of the development of RTAs as a whole, formulate differentiated development strategies for different development stages, and focus on strengthening the construction of important border areas and important behind-border areas to facilitate the establishment of deep RTAs.

Besides these implications, there are also some limitations in our empirical research. This study is an empirical analysis for 143 countries around the world, not a specific study. Future studies could focus on a particular country or an economic organization to examine the nexus. Meanwhile, this study mainly analyzes the impact mechanisms of deep free trade agreements on residents’ health from the perspectives of employment and environment, but there may be other mediating factors. Future studies can explore the mediating mechanisms of deep RTAs on residents’ health more comprehensively and test them empirically. In addition, both residents’ health and RTAs are comprehensive concepts. For example, residents’ health includes different aspects such as happiness, life expectancy, and death rate, etc., and RTAs includes different terms such as export taxes, investment, and intellectual property rights, etc. Therefore, future studies can focus on an aspect of residents’ health or a particular term of RTAs to analyze the impact of deep RTAs on different aspects of residents’ health and the impact of the depth of different terms in RTAs on residents’ health.

## Figures and Tables

**Table 1 ijerph-19-14409-t001:** Value of term.

Term	Value	Term	Value	Term	Value
Rules of origin	0–38	Environment	0–55	SPS	0–59
Trade facilitation	0–52	Export taxes	0–55	TBT	0–34
Competition policies	0–35	Labor market	0–23	STE	0–61
Intellectual property	0–136	Anti-dumping	0–14	Services	0–64
Public procurement	0–100	State subsidies	0–44	Migration	0–36
Countervailing duties	0–8	Movement of capitals	0–136	Investment	0–58

**Table 2 ijerph-19-14409-t002:** Variables definition and sources.

Variables	Symbols	Definitions	Sources
Residents’ health	PHI	Personal Health Index	Gallup World Poll Database
Depth of trade agreements	Depth	Standardized values of the scores of 1028 sub-terms	World Bank Trade Agreement Content Database
Residents’ income	Lnincom	Logarithmic values of total annual household net income (current USD)	Gallup World Poll Database
Residents’ education level	Education	Basic education (1), vocational education (2), higher education (3)	Gallup World Poll Database
Residents’ age	Age	The actual age of residents	Gallup World Poll Database
Resident’ gender	Male	Male (1), female (0)	Gallup World Poll Database
Residents’ marital status	Marriage	Married (1), unmarried (0)	Gallup World Poll Database
Residents’ residential area	Urban	City or suburb (1), rural area or small town (0)	Gallup World Poll Database
GDP per capita	LnGDPpc	Logarithmic values of GDP per capita (constant 2015 USD)	World Bank Development Indicators Database
Health expenditure per capita	Lnhepc	Logarithmic values of the health care expenditure per capita (PPP, current USD)	World Bank Development Indicators Database
Foreign trade dependence	FTD	the proportion of import and export trade in the gross national product	World Bank Development Indicators Database.
Average annual PM2.5 exposure.	Pollution	PM2.5 air pollution, mean annual exposure (micrograms per cubic meter)	World Bank Development Indicators Database.
Labor force participation rate	Laborpar	Labor force participation rate, total (% of population ages 15–64)	World Bank Development Indicators Database.

**Table 3 ijerph-19-14409-t003:** Descriptive statistics.

Variables	Obs	Mean	Std. Dev	Min	Max
PHI	786,040	70.2256	28.2378	0.0000	100.0000
Depth	786,040	−0.5487	1.0090	−1.5423	1.6276
Lnincom	786,040	9.3068	1.3215	−2.5257	20.6157
Education	786,040	1.8869	0.6996	1.0000	3.0000
Age	786,040	42.0853	11.3180	25.0000	64.0000
Male	786,040	0.4595	0.4984	0.0000	1.0000
Marriage	786,040	0.7114	0.4531	0.0000	1.0000
Urban	786,040	0.4230	0.4940	0.0000	1.0000
LnGDPpc	786,040	8.7833	1.4080	5.3572	11.7254
Lnhepc	786,040	6.5804	1.2867	3.9504	8.7995
FTD	786,040	84.5028	53.1202	19.4600	416.3900
Pollution	719,491	32.0431	22.4343	5.8613	100.7844
Laborpar	786,040	68.0740	10.3809	41.5300	89.9800

**Table 4 ijerph-19-14409-t004:** Benchmark results.

Variables	(1)	(2)	(3)
Depth	2.169 *** (0.764)	2.306 *** (0.705)	1.987 *** (0.645)
Lnincom		4.457 *** (0.179)	4.458 *** (0.181)
Education		2.981 *** (0.202)	2.978 *** (0.203)
Age		−0.294 *** (0.0148)	−0.294 *** (0.0149)
Male		3.256 *** (0.244)	3.247 *** (0.244)
Marriage		2.172 *** (0.153)	2.166 *** (0.153)
Urban		−0.654 *** (0.219)	−0.659 *** (0.220)
LnGDPpc			−1.537 (1.439)
Lnhepc			0.0399 ** (0.0178)
FTD			2.751 * (1.574)
Constant	71.58 *** (3.922)	37.31 *** (3.561)	28.87 ** (11.18)
Country fixed effect	YES	YES	YES
Year fixed effect	YES	YES	YES
Country linear time trend	YES	YES	YES
Observations	786,040	786,040	786,040
R-squared	0.048	0.099	0.099

Note: Standard errors in parentheses; * *p* < 0.10, ** *p* < 0.05, *** *p* < 0.01.

**Table 5 ijerph-19-14409-t005:** Robustness test results.

Variables	(1)	(2)	(3)	(4)
Averagedepth	0.852 *** (0.288)			
Depth		1.609 ** (0.692)	0.125 *** (0.0432)	2.093 *** (0.629)
Constant	27.88 ** (11.41)	34.86 (271.8)		33.55 *** (11.91)
Control variables	YES	YES	YES	YES
Country fixed effect	YES	YES	YES	YES
Year fixed effect	YES	YES	YES	YES
Country linear time trend	YES	YES	YES	YES
Observations	786,040	541	786,040	947,650
R-squared	0.099	0.334	0.029	0.097

Note: Standard errors in parentheses; ** *p* < 0.05, *** *p* < 0.01.

**Table 6 ijerph-19-14409-t006:** Endogeneity test results.

Variables	(1)	(2)	(3)
${\mathrm{Depth}}^{\mathrm{IV}}$	3.398 *** (1.164)	3.805 *** (1.110)	3.411 *** (1.057)
Lnincom		4.459 *** (0.178)	4.460 *** (0.180)
Education		2.979 *** (0.201)	2.977 *** (0.202)
Age		−0.294 *** (0.0148)	−0.294 *** (0.0148)
Male		3.254 *** (0.243)	3.245 *** (0.244)
Marriage		2.176 *** (0.152)	2.171 *** (0.153)
Urban		−0.661 *** (0.219)	−0.665 *** (0.219)
LnGDPpc			−1.406 (1.448)
Lnhepc			2.635 * (1.556)
FTD			0.0356 ** (0.0177)
Control variables	73.61 *** (4.251)	39.76 *** (3.872)	31.33 *** (11.05)
Country fixed effect	YES	YES	YES
Year fixed effect	YES	YES	YES
Country linear time trend	YES	YES	YES
Observations	786,040	786,040	786,040
Kleibergen–Paap rk LM	12.859 [0.000]	12.856 [0.000]	12.691 [0.000]
Kleibergen–Paap rk Wald F	43.656 {16.38}	43.651 {16.38}	41.590 {16.38}
R-squared	0.048	0.099	0.099

Note: Standard errors in parentheses; * *p* < 0.10, ** *p* < 0.05, *** *p* < 0.01; the value in square brackets is the *p* value of the corresponding statistic; the value in curly brackets is the critical value at the 10% level of the Stock–Yogo test.

**Table 7 ijerph-19-14409-t007:** Mechanism test results.

Variables	(1)	(2)	(3)	(4)
	PHI	Laborpar	Pollution	PHI
Depth	1.987 *** (0.645)	0.644 * (0.355)	−1.116 *** (0.396)	1.849 *** (0.600)
Laborpar				0.474 *** (0.134)
Pollution				−0.137 * (0.0721)
Constant	28.87 ** (11.18)	49.58 *** (5.255)	123.2 *** (17.26)	17.60 (14.16)
Control variables	YES	YES	YES	YES
Country fixed effect	YES	YES	YES	YES
Year fixed effect	YES	YES	YES	YES
Country linear time trend	YES	YES	YES	YES
Observations	786,040	786,040	719,491	719,491
R-squared	0.099	0.988	0.988	0.101

Note: Standard errors in parentheses; * *p* < 0.10, ** *p* < 0.05, *** *p* < 0.01.

**Table 8 ijerph-19-14409-t008:** Results based on different terms and different income levels.

Variables	(1)	(2)	(3)	(4)
	“WTO+” Terms	“WTO-X” Terms	Low-Income and Middle-Income Countries	High-Income Countries
LnWTO+	2.057 *** (0.680)			
LnWTO-X		1.950 *** (0.590)		
Depth			4.843 *** (1.484)	0.426 (0.436)
Constant	27.53 ** (11.23)	28.66 ** (11.13)	13.26 (16.52)	29.83 * (15.35)
Control variables	YES	YES	YES	YES
Country fixed effect	YES	YES	YES	YES
Year fixed effect	YES	YES	YES	YES
Country linear time trend	YES	YES	YES	YES
Observations	786,040	786,040	293,767	492,273
R-squared	0.099	0.099	0.097	0.100

Note: Standard errors in parentheses; * *p* < 0.10, ** *p* < 0.05, *** *p* < 0.01.

**Table 9 ijerph-19-14409-t009:** Results based on different types of residents.

Variables	(1)	(2)	(3)	(4)
	Urban Residents	Rural Residents	Basic Education and Vocational Education Residents	Higher Education Residents
Depth	0.972 * (0.569)	2.690 *** (0.767)	2.126 *** (0.728)	0.414 (0.578)
Constant	36.79 *** (10.35)	16.68 (14.80)	28.03 ** (11.92)	39.13 *** (12.10)
Control variables	YES	YES	YES	YES
Country fixed effect	YES	YES	YES	YES
Year fixed effect	YES	YES	YES	YES
Country linear time trend	YES	YES	YES	YES
Observations	332,472	453,568	633,099	152,941
R-squared	0.097	0.103	0.098	0.055

Note: Standard errors in parentheses; * *p* < 0.10, ** *p* < 0.05, *** *p* < 0.01.

## Data Availability

The datasets used and/or analyzed during the current study are available from the corresponding author on reasonable request.

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
