# Peer review of "Do Deep Regional Trade Agreements Improve Residents’ Health? A Cross-Country Study"

_ijerph, 2022, doi:10.3390/ijerph192114409_

Round 1

Reviewer 1 Report

The article was prepared with great detail.
In the introduction and conclusion, ensure that the material is properly cited.
Well-chosen material to analyze and prove research hypotheses. Proper presentation of research results and well drawn conclusions.

Article is well prepared and structure. Minor adjustment with citation are needed. For exapmle in line 27-29 the information from World Bank are presented without showing the source.

After the small changes article is ready to be published.

Author Response

    Thank you for your valuable suggestions, which are benefit for improving the quality of this paper. We have revised and supplemented the manuscript. Replies to related suggestions are as follows:

    We have supplemented the original source of the information in line 29-31. The specific supplementary contents are as follows: "WTO Regional Trade Agreements Database". Meanwhile, we also provide specific URLs of the data source. The specific URLs is as follows: http://rtais.wto.org/UI/PublicMaintainRTAHome.aspx.

    Thank you again for your approval and recognition of this paper.

Reviewer 2 Report

This article is of great interest as it deals with a relevant and topical issue. However, some improvements should be made:

In my view "Evidence from 143 Countries" in the title does not add value, could it be reworded? 

In line 82 it says: "Meanwhile, some scholars have used qualitative or dummy variable analysis methods to analyse the impact of regional trade agreements on residents' health". Could you cite some of the most important studies? 

In line 84 it says: "Few studies have examined the impact of deep regional trade agreements signed around the world on residents' health from the perspective of depth of terms, and there are also few studies that analyse in depth the mechanisms by which deep free trade agreements affect residents' health". What does this article add value to these studies? 

The Theoretical Framework does not adequately support existing theory. The number of references needs to be increased. 

Line 241 reads: "The sample period ranges from 2009 to 2017". Why did you choose that time period?

Furthermore, in this same paragraph (line 241 and 242): "and the sample covers 786,040 residents in 143 countries, including mainly some major developed countries such as the United States, United Kingdom, Germany, France, Japan, and some major developing countries such as China, Malaysia, Brazil, India and the Russian Federation" Why did you choose this sample? Do these countries have something in common for the methodological development of this study? 

A Discussion of results section is needed 

In line 529 it is argued that: "Future studies can explore the mediating mechanisms of deep free trade agreements on residents' health more comprehensively and test them empirically". Does only this future line of research exist? Could you provide some more? 

Author Response

    Thank you for your valuable questions and suggestions, which are benefit for improving the quality of this paper. We have revised and supplemented the manuscript. Replies to related questions and suggestions are as follows:

  1. We have modified the title. The new title is as follows: " Do Deep Regional Trade Agreements Improve Residents' Health? A Cross-Country Study".
  2. We have cited some of the most important studies which used qualitative or dummy variable analysis methods to analyze the impact of regional trade agreements on residents' health. The specific cited contents are as follows: "For example, Stiglitz [34] qualitatively analyzed the negative effects of traditional RTAs on the health of residents in developing countries; Venkatamaran & Stevens [35] analyzed the positive effects of traditional FTAs on health outcomes using the dummy variable method based on the sample of countries that had signed FTAs with the United States or the European Union, etc."
  3. In line 92-95, We are trying to express that these studies don't yet exist in the academic community, but there is a misrepresentation. Therefore, we have modified the statements. The modified statements are as follows: "However, there are no studies examining the impact of deep regional trade agreements signed around the world on residents' health from the perspective of terms depth, and there are also no studies deeply analyzing the mechanisms by which deep free trade agreements affect residents' health." In addition, in order to highlight the marginal contributions of this paper compared to existing studies, we have supplemented the following contents: "Compared with existing studies, the contributions of this study are as follows."
  4. We have enriched the theoretical framework and increased the number of references. The specific supplementary contents are as follows:

(1) "reduce the trade costs that enterprises need to pay for export, including fixed trade costs such as market entry and information exchange, etc., as well as variable trade costs such as product transportation and trial communication, etc., and reduce trade diversion while promoting trade creation [3]. "

(2) "and specialized health investments such as: travel, vacations, fitness, exercise, etc. [41]. "

(3) "and improve the subjective health level of residents' self-assessment [41]".

(4) "Therefore, deep RTAs can form a kind of environmental regulation for domestic enterprises in each country. The environmental regulation will force enterprises to shift their production processes to cleaner ones and force them to abandon the production of polluting products [42]. This greatly promotes cleaner production of enterprises and the import of cleaner intermediate products, which ultimately reduces pollutants that are harmful to residents' health, such as PM2.5, and improves domestic environmental quality. "

(5) "Meanwhile, environmental pollution will increase the health expenditure of residents [45], which will indirectly reduce the income of residents and increase the stress of their lives, and eventually leads to the decline of residents' health. "

(6) "and reduce the pressure on residents' health expenses".

  1. We chose the sample period with two main considerations. First, the dependent variable of this paper is residents' health, which is obtained from the Gallup World Poll Database, and the coverage of this database is from 2009 to 2019. Second, the independent variable of this paper is the depth of regional trade agreements, which is measured based on the World Bank Trade Agreement Content Database, and the coverage of this database is from 1949 to 2017. Therefore, on the basis of considering the data availability of the dependent and independent variables, the sample period of this paper ranges from 2009 to 2017.
  2. The main reason for choosing 143 countries as our sample is that the Gallup World Poll database contains data on residents' health only for 143 countries, but these countries cover major developed and developing countries worldwide. Meanwhile, we chose to list the major developed countries, such as the United States, the United Kingdom, Germany, France, Japan, and the major developing countries, such as China, Malaysia, Brazil, India, the Russian Federation, in order to illustrate the comprehensiveness of the sample and the broad applicability of the empirical results.
  3. We not only reported the results of the benchmark empirical, robustness test, endogeneity test, mechanism analysis, and heterogeneity analysis in Section 4, but also provide the detailed and systematic discussions of each type of results. To better indicate that the regression results have been discussed in this paper, we change the title of section 4 to "Results and Discussions".
  4. We have supplemented the future research directions. The specific supplementary contents are as follows: "In addition, both residents' health and RTAs are comprehensive concepts. For example, residents' health includes different aspects such as happiness, life expectancy, and death rate, etc., and RTAs includes different terms such as export taxes, investment, and intellectual property rights, etc. Therefore, future studies can focus on an aspect of residents' health or a particular term of RTAs to analyze the impact of deep RTAs on different aspects of residents' health and the impact of the depth of different terms in RTAs on residents' health."

    Thank you again for your questions and suggestions.

Reviewer 3 Report

The manuscript titled “Do Deep Regional Trade Agreements Improve Residents' Health? Evidence from 143 Countries” presents a new approach to studying factors influencing population health across the World. The manuscript suits the scope of the International Journal of Environmental Research and Public Health.

I can see the following issues that should be resolved before this research paper is published.

The abstract needs to be revised. It is incomplete as it lacks the research methods employed.

The research is based on unique individual-level Gallup World Polls data. The “Materials and Methods” section should inform readers of how the Authors accessed Gallup data for their research. Did the Authors obtain the data by submitting a request to Gallup directly or otherwise?

Sub-section 3.2.1. too briefly describes the dependent variable named by the Authors “residents’ health index” which corresponds to Gallup’s “The Personal Health Index” (PHI). Firstly, it is worth referring to the Gallup publication on the World Poll methodology (e.g. Gallup, 2021: Worldwide Research Methodology and Codebook. Gallup, Washington D.C.). Second, it would be valuable for readers to be aware of the reliability of this index. Readers should know that PHI is a subjective measure as well as what it measures, namely “perceptions of one’s own health, both physical and mental and incidence of pain, sadness, and worry”. Moreover, I suggest that the Authors list the questions related to the Personal Health Index that Gallup poses in the GWP questionnaire.

Sub-sections 3.2.2. and 3.2.3. The dependent variables and control variables would be more readable if they were presented and fully characterized (including their acronyms and sources) in tabular form. More scrutiny should be paid to describing the variables in detail. A selected example includes (219-220): “residents' income, measured by a total annual household income logarithm” – What specific income? Gross, net, from what sources? Currency unit? In current prices or in constant prices? Other examples: GDP per capita (PPP or not?); “health care expenditure per capita, measured by purchasing power parity” (constant or current prices?); “foreign trade dependence, measured by the proportion of import and export trade in the gross national product (for sure GNP and not GDP?), etc.

In my opinion, present subsections 3.1.1. and 3.1.2. should follow sub-sections presenting variables and sources. When giving the models' specifications, the Authors name and use these variables which are later described in the following subsections. Shouldn't the order be reversed?

Sub-section 3.3. It is very important to connect all specific data to their sources of origin and the Authors have done so in this manuscript. Regrettably, the Authors provide neither in-text references (citations) to the data sources nor database URLs (e.g. World Bank datasets including the Deep Trade Agreements database).

Results: The Authors are recommended to check carefully and reconsider the interpretation of their results. For instance, the dependent variable PHI is expressed in points (0-100) as well as variable ????ℎ is expressed in points. Consequently, it seems that the statement (275-277) “the Column (3) is 1.987, which means that a 1 percent increase in the depth of regional trade agreements will improve residents' health by 1.987 percent” is not correct. Benchmark Regression Model is linear regression, so the change in the independent variable is 1 point (not percent), as well as change in the independent variable, is in points, not percent.

Please also consider a few minor editorial comments provided below.

Overall, this manuscript is written exceptionally well. The sections generally follow clearly. I suggest reconsidering the structure of section 3, as shown in my previous comments.

170-171: “residents' gender (????), residents' marital status (????????), and residents' residential area (?????)” – only variable names are given without subscripts ??” as opposed to the case of other variables (see ?????)

243: “United States, United Kingdom” – correctly: the United States, the United Kingdom

256-257: “health index value is 70.2256, indicating a fair residents' health status”. In fact, PHI is about an individual's perception of health, not about objective health status.

 I propose using abbreviations (e.g. regional trade agreements - RTAs) instead of repeating phrases frequently throughout the document. As a result, the text will be shortened and will be easier to read.

Author Response

    Thank you for your valuable questions and suggestions, which are benefit for improving the quality of this paper. We have revised and supplemented the manuscript. Replies to related questions and suggestions are as follows:

  1. We have modified the abstract. First, we supplemented a description of the research methods used in this paper, and the specific supplementary contents are as follows: "by using the combination of fixed effects and stepwise regression". Second, we further enriched the study context in the abstract, and the specific enrichment contents are as follows: "traditional RTAs based on border terms continue to shift to deep RTAs based on the high level of border terms and a series of post-border terms".
  2. In the section of "Data Sources and Statistical Characteristics", We have supplemented some contents to inform readers of how the authors accessed the Gallup data for the research. The specific supplementary contents are as follows: "which can be obtained by submitting a request to the Gallup".
  3. In the section of "Dependent Variable", We have supplemented the description of residents’ health. The specific supplementary contents are as follows: "To be specific, the Personal Health Index is a subjective measure, which represent the perceptions of one’s own health, both physical and mental and incidence of pain, sadness, and worry. And the Personal Health Index is measured based on the Gallup questionnaire, which includes a standard set of questions related to personal health. For example, have you tried to get any information about medicine, disease, or health in the past 30 days? Would you like to know more about medicine, disease, or health? How much do you trust medical and health advice from medical workers, such as doctors and nurses, in this country? etc., and the results of Gallup questionnaire can represent more than 98% of the world’s adult population."
  4. We have added a new table, Table 2 "Variables definition and sources", which presented the information about the symbols, definitions, and data sources of the independent, dependent, and control variables.
  5. We have reordered the three subsections in Section3. Specifically, we have placed the original subsection 3.1 after subsections 3.2 and 3.3. Then we put the original 3.2 and 3.3 as 3.1 and 3.2, respectively.
  6. In the section of "Data Sources and Statistical Characteristics ", We have supplemented the URLs of the World Bank Trade Agreement Content Database and the World Bank Development Indicators Database.
  7. We have revised the interpretation of the results. The revised explanations of the results are as follows: "which means that one point increase in the depth of regional trade agreements will improve residents' health by 1.987 points."
  8. We have supplemented the subscripts “??” of residents' gender (????), residents' marital status (????????), and residents' residential area (?????).
  9. We have revised "United States" and "United Kingdom" to "the United States" and "the United Kingdom" respectively.
  10. We have deleted the following inappropriate statements: "indicating a fair residents' health status requiring improvement."
  11. We have changed the "regional trade agreements" in this paper to "RTAs".

    Thank you again for your questions and suggestions.

Round 2

Reviewer 2 Report

The manuscript has improved considerably. Congratulations!